# Deep Learning-Based Anomaly Detection in Industrial Images: Evaluation and Comparison of Modern Methods

Anonymous Full Paper
Submission 14

## Abstract

Unsupervised anomaly detection (AD) plays a key role in industrial quality control, enabling automated early detection of defective products without the need for collecting extensive amounts of labeled anomalous training data. However, practitioners and researchers are challenged by choosing suitable AD models in realistic use cases. In this work, we therefore evaluate three modern unsupervised AD models, namely *EfficientAD*, *PatchCore*, and *FastFlow*, on (1) the well-established *MVTec AD* dataset and (2) the new real-world *Rubber Mats* dataset, which contain subtle and diverse defects. In a design-science-oriented experimental study, the AD models selected are benchmarked. Further, the impact of higher image resolutions is evaluated, and generalization in multi-class training across categories of instances is examined. All models achieve very high overall accuracy but differ in strengths: *EfficientAD-M* balances accuracy and efficiency best, *PatchCore* benefits from higher resolutions but at the cost of efficiency, and *FastFlow* generalizes well with stable accuracy. Our findings emphasize the importance of selecting models based on application-specific needs. The results, supported by an open-source benchmark implementation, offer practical insights for deploying visual AD systems in real production environments.

## 1 Introduction

Visual anomaly detection (AD) is a critical task in industrial quality assurance, enabling the automated identification of defective products during production. Early and accurate detection reduces waste, improves efficiency, and ensures consistent product quality [1]. Traditional manual inspection, however, is labor-intensive, error-prone, and difficult to scale [1, 2]. Recent advances in deep learning have transformed AD, with modern methods achieving state-of-the-art accuracy [2–4].

Despite this progress, selecting the most fitting AD model for a specific industrial scenario remains challenging. In industrial production, defects manifest as subtle deviations, making fine-grained defect detection difficult. At the same time, models must satisfy both accuracy and efficiency requirements to be relevant in real-world deployments [2, 4]. Moreover, unseen defects can occur at any time, while collecting and labeling large amounts of anomalous data is costly and time-consuming [2, 5].

Unsupervised methods address these challenges by relying exclusively on normal data for model training, thereby minimizing annotation effort [2]. However, these approaches differ in terms of accuracy, inference efficiency, and generalization capability, although model performance varies between datasets, defect types, and image resolutions. Consequently, model selection in practice is highly application-dependent [6]. This leads us to the central research question of this work:

"*How can deep learning-based, unsupervised anomaly detection models for industrial image classification be best selected in practice?*"

Therefore, we present a systematic benchmark of three representative, modern unsupervised AD methods: *EfficientAD*, *PatchCore*, and *FastFlow*. Beyond classification accuracy, we analyze inference efficiency, the influence of image resolutions, and model generalization in a multi-class setting. First, we evaluate all methods under standard resolution to establish baseline performance in terms of accuracy and efficiency. Second, we analyze the impact of image resolution on model performance, with a particular focus on small, fine-grained defects. Finally, we extend the evaluation to a multi-class benchmark, where models are trained jointly across dataset categories, to assess their ability to generalize to more heterogeneous industrial scenarios. In addition to the *MVTec AD* dataset, we include a new real-world *Rubber Mats* dataset, which contains more complex textures and subtle defects, to provide a broader assessment of robustness in realistic industrial conditions.

Our contributions are therefore threefold:

1. A comprehensive benchmark of modern unsupervised AD methods on *MVTec AD* and the new *Rubber Mats* dataset, including comparisons of classification accuracy and efficiency metrics.

2. An in-depth analysis of the effect of image resolutions on anomaly detection accuracy and efficiency, highlighting model-specific trade-offs.

3. An evaluation of multi-class AD of the selected methods, providing additional insights into model generalization beyond the commonly studied single-class setting.

This work aims to support practitioners and researchers in choosing appropriate AD models by offering a comparative analysis grounded in realistic use cases. Our findings indicate that no universally superior model exists. Instead, trade-offs depend on dataset complexity, defect characteristics, and deployment requirements. To facilitate reproducibility and further experimentation, the implementation details are released as open source.

Finally, this research is positioned within a design-oriented framework following the Design-Science-Research Methodology (DSRM) [7]. Thus, section 2 introduces the foundations of AD, unsupervised learning, and AD-specific approaches. Thereafter, section 3 details the research design and experimental setup. Section 4 presents the empirical results, analyzing model performance under standard conditions, varying image resolutions, and a multi-class setting. Lastly, section 5 discusses the findings and concludes the research.

## 2 Background

AD refers to the identification of subtle deviations from the expected norm. In *industrial manufacturing*, defects are often rare, unpredictable, and localized to small regions within otherwise normal images [8]. As a result, *unsupervised AD* has become particularly important, since it eliminates the need for extensive labeled training data covering diverse defect types [2, 5].

Unsupervised models are trained exclusively on normal data and detect anomalies by identifying deviations from learned normal representations during inference [8]. Typically, anomaly scores are computed at the pixel level and aggregated into image-level scores, which are then converted into binary classification decisions via thresholding [9, 10].

Among modern approaches, the following model paradigms have proven especially effective in recent benchmarks: *PatchCore* [11] is a memory bank-based method that extracts patch-level features from normal images and stores a representative subset in its memory bank, using greedy coreset subsampling for lower inference cost and higher performance. At inference, test features are compared against stored ones using K nearest neighbor search to detect deviations. By combining local patch features with strong *ImageNet* [12] pretrained feature extractors, *PatchCore* achieves high accuracy and robust generalization.

The teacher-student-based *EfficientAD* [4] trains a lightweight student network to mimic the extracted feature representation of a pretrained teacher model on normal data. During inference, anomalies are identified from deviations of extracted features between the teacher and student model. *EfficientAD* further enhances robustness to logical anomalies, such as missing or misplaced components, by additionally integrating a global-context autoencoder.

Finally, distribution-map-based methods such as *FastFlow* [13] learn to estimate the probability distribution of normal features. During inference, anomalies are identified when features deviate from this learned distribution. *FastFlow* supports fully convolutional, efficient inference and captures both local and global feature distributions by utilizing two-dimensional normalizing flows.

Several benchmarking studies highlight strengths and limitations in these methods. Xie et al. [14] find that feature-embedding-based methods, especially memory bank and teacher-student architectures, generally outperform reconstruction-based approaches for anomaly classification tasks, though no single method dominates across all datasets. Furthermore, methods utilizing strong pretrained models as feature extractors, such as *ResNet* [15], often achieve better performance than methods without [5].

Zhang et al. [10] observe that multi-class training can degrade performance in some methods, while *PatchCore* shows stronger robustness. Similarly, Han et al. [16] conduct an extensive benchmark across 50+ datasets and report that model performance is highly scenario-dependent, underscoring the need for application-specific evaluation.

Efficiency is another key factor. Batzner et al. [4] note that some methods prioritize accuracy while compromising inference speed, which limits real-time industrial adoption. In addition, they show that *EfficientAD* achieves a strong balance of high accuracy and efficiency. Rolih et al. [6] further demonstrate that higher image resolutions may improve accuracy for some models, such as *PatchCore*, but can also increase latency or even degrade performance, depending on the model.

Despite progress in unsupervised AD, important research gaps remain. Benchmarks often focus on single-class training and standard resolutions, overlooking how higher resolutions influence defect detection. In industrial settings, however, fine-grained defects and large-resolution inputs are common, while excessive downscaling risks losing critical details. Moreover, real-world production may involve multi-class environments, but model generalization and scalability under such conditions are rarely evaluated. Additionally, aggregate dataset-level results dominate reporting, while instance-level insights (e.g., defect characteristics) are equally important for practical model selection. Limitations are compounded by the scarcity of realistic and high-quality datasets, as many existing ones are synthetic or not collected from production environments [2]. Finally, focusing on image-level classification accuracy remains highly relevant in practice due to its reduced labeling effort compared to pixel-level anomaly localization.

To address these gaps, a comprehensive and scenario-aware benchmark was realized in this research that jointly considers model performance by multiple indicators, such as accuracy, efficiency, sensitivity to image resolution, and multi-class robustness. The presented study thus aims to sensitize the practical deployment of visual AD systems in industrial applications.

## 3 Methodology

### 3.1 Model selection

We evaluate three unsupervised AD models, namely (1) *PatchCore*, (2) *EfficientAD*, and (3) *Fastflow*, each representing a different model architecture paradigm. These models were selected based on their strong performance in recent benchmarks [2, 4, 11, 13, 14, 17] and their availability in the *Anomalib* library [18], which offers standardized implementations of state-of-the-art AD methods, enabling reproducibility and consistent evaluation. To better explore accuracy-efficiency trade-offs, we additionally considered lightweight and high-performance model variants where applicable, namely (1a) *PatchCore-10%* and (1b) *PatchCore-1%*, as well as (2a) *EfficientAD-S* and (2b) *EfficientAD-M*. More precisely, *PatchCore-1%* differs by a larger *Wide-ResNet101* backbone and a smaller 1% subsampling ratio, and *EfficientAD-M* is distinguished by more convolutional layers than its S-variant. The detailed model hyperparameters are shown in Tab. A.1.

### 3.2 Datasets

Experiments were conducted on two datasets, namely (1) the *MVTec AD* dataset [8] and (2) the *Rubber Mats* dataset. The training set for both consists of solely normal images, while the test set includes normal and anomalous images. *MVTec AD* is a widely used industrial benchmark dataset with 15 categories, particularly 10 objects and 5 textures, with many diverse defect types and sizes.

*Rubber Mats* is a real-world dataset collected from an industrial manufacturing process. It consists of 198 training and 54 high-resolution (1440×1080 pixels) test images featuring subtle texture defects, examples of which are shown in Fig. 1. Compared to *MVTec AD*, *Rubber Mats* exhibits greater variability in normal samples and more fine-grained anomalies, making it particularly suitable for assessing sensitivity to nuanced, realistic defects. Each dataset category can be treated as a separate training task to reflect isolated industrial use cases.

To analyze the behavior of the model, categories have been grouped by defect characteristics. Specifically, we distinguished between *smaller-defect categories*, which include *Rubber Mats* and *MVTec AD*'s

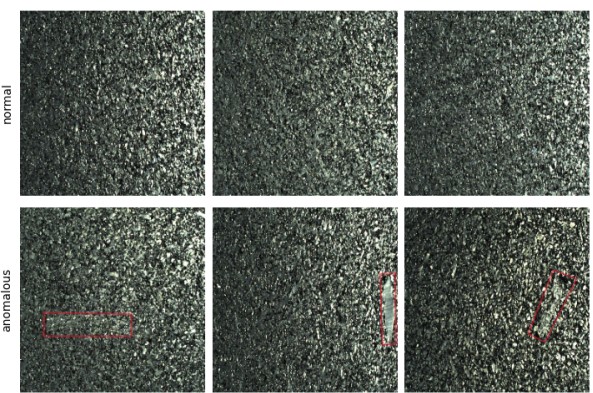

**Figure 1.** Selected images from the *Rubber Mats* dataset, showing normal and anomalous samples. Defects are annotated with red bounding boxes in the anomalous cases.

categories *grid*, *capsule*, *pill*, and *screw*, as well as *larger-defect categories*, comprising the remaining *MVTec AD* classes. The grouping is based on the relative defect size between categories (see Fig. A.1), as well as visual inspection. In addition, categories can be separated into textures and objects, enabling evaluation across both contexts.

### 3.3 Metrics

We evaluated the model performance in terms of *accuracy* and *efficiency*. Classification accuracy is measured using AU-ROC and AU-PR, which are threshold-independent. AU-ROC quantifies detection of and separation between normal and anomalous instances, while AU-PR emphasizes correct detection of true anomalies and is particularly relevant in imbalanced datasets [8, 19].

Inference *efficiency* is quantified by latency and throughput, following the proposed protocol from Batzner et al. [4]. Latency is measured by average inference time per single image, and throughput is defined as images per second in batch-wise evaluation. Each efficiency metric is computed from 1000 inference runs, excluding warm-up iterations to ensure stable measurements.

### 3.4 Experimental Setup

We designed three complementary experiments to evaluate accuracy, efficiency, and generalization.

*Standard resolution benchmark:* All models have been trained separately on each dataset category using a 256×256 pixel input resolution. This single-class setup reflects common practice in AD research and serves as the baseline for comparison.

*Image resolution analysis:* To assess how input resolution affects performance, especially for smaller-defect categories, all models are additionally evaluated at higher resolutions of 320×320 and 448×448

**Table 1.** Classification accuracy (AU-ROC/AU-PR, [%]) of all evaluated models on *MVTec AD* and *Rubber Mats* at the standard image resolution of 256×256.

| AD Model Category | PatchCore-10 | PatchCore-1 | EfficientAD-S | EfficientAD-M | FastFlow |
|---|---|---|---|---|---|
| Carpet | 99.2/99.8 | 98.6/99.6 | **100/100** | 99.9/**100** | 99.8/**100** |
| Grid | 98.3/99.5 | 98.1/99.5 | **100/100** | **100/100** | **100/100** |
| Leather | 100/100 | 100/100 | 100/100 | 100/100 | 100/100 |
| Tile | 98.7/99.6 | **100/100** | **100/100** | **100/100** | **100/100** |
| Wood | 98.8/99.7 | 98.7/99.6 | 99.6/99.9 | **100/100** | 99.8/99.9 |
| Bottle | 100/100 | 100/100 | 100/100 | 100/100 | 100/100 |
| Cable | 98.7/99.2 | **99.5/99.7** | 96.2/97.6 | 98.0/98.7 | 96.9/97.9 |
| Capsule | 97.7/99.5 | 98.6/99.7 | 97.6/99.5 | 96.2/99.1 | **99.0/99.8** |
| Hazelnut | **100/100** | **100/100** | 95.4/97.5 | 95.4/97.6 | 99.9/99.9 |
| Metal nut | 99.8/**100** | 100/100 | 99.1/99.8 | 99.4/99.9 | **100/100** |
| Pill | 94.3/98.9 | 94.7/98.9 | 98.9/99.8 | 97.8/99.6 | **99.2/99.9** |
| Screw | 98.1/99.3 | 96.3/98.7 | 98.2/99.4 | **98.7/99.6** | 88.4/94.0 |
| Toothbrush | **100/100** | 94.4/97.6 | 99.5/99.8 | 99.2/99.7 | 90.8/95.7 |
| Transistor | **100/100** | **100/100** | 98.4/97.5 | 99.6/99.4 | 99.4/99.1 |
| Zipper | 99.1/99.8 | 98.7/99.6 | 99.6/**99.9** | 99.7/99.9 | 99.6/99.8 |
| MVTecAD | 98.8/**99.7** | 98.5/99.5 | 98.8/99.4 | **98.9**/99.6 | 98.2/99.1 |
| RubberM | 98.1/98.6 | **99.0/99.2** | 92.6/94.3 | **99.0/99.2** | 98.1/98.5 |
| **Mean** | 98.8/**99.6** | 98.5/99.5 | 98.4/99.1 | **98.9**/99.5 | 98.2/99.0 |

**Table 2.** Inference efficiency (latency [ms]/throughput [img/s]) and mean accuracy (AU-ROC/AU-PR, [%]) of evaluated models for image resolution 256×256.

| Metrics Selected AD Model | Classification Accuracies | | | Efficiencies | |
|---|---|---|---|---|---|
| | MVTecAD | RubberM | Mean | Latency | Throughput |
| PatchCore-10 | 98.8/**99.7** | 98.1/98.6 | 98.8/**99.6** | 12.6 | 201 |
| PatchCore-1 | 98.5/99.5 | **99.0/99.2** | 98.5/99.5 | 17.1 | 408 |
| EfficientAD-S | 98.8/99.4 | 92.6/94.3 | 98.4/99.1 | **4.5** | **630** |
| EfficientAD-M | **98.9**/99.6 | **99.0/99.2** | **98.9**/99.5 | 6.8 | 287 |
| FastFlow | 98.2/99.1 | 98.1/98.5 | 98.2/99.0 | 29.7 | 376 |

pixels. Leveraging *PatchCore*'s coreset subsampling to maintain efficiency at varying scales, it is also evaluated at 128×128 and 512×512 pixels, and *PatchCore-1%* is used for higher image scaling beyond standard resolution.

*Multi-class generalization:* To examine robustness in multi-class settings, models are trained jointly across grouped *MVTec AD* categories of textures and objects at the standard resolution. This setup reflects industrial scenarios where one model must generalize across multiple classes and defect types simultaneously.

All models are evaluated under consistent image resolutions and recommended hyperparameters to ensure comparability, while all experiments are implemented using *Anomalib* and executed on an NVIDIA A100 GPU. Full implementation details are available in our open-source repository: [BlindedRepo].

# 4 Results

## 4.1 Standard Resolution Benchmark

We first evaluated all models at the standard input resolution of 256×256 pixels. Results are reported in terms of AU-ROC and AU-PR for anomaly classification accuracy, along with inference efficiency. Selected examples for true positive detections of every model can be found in the appendix at Fig. A.2. Corresponding misclassified test images for every model can be found at Fig. A.3.

### 4.1.1 Accuracy Results

Table 1 summarizes the accuracy results. Across MVTec AD, all models achieve high AU-ROC and AU-PR scores, with EfficientAD-M slightly leading with 98.9% AU-ROC and 99.6% AU-PR, closely followed by PatchCore-10% and EfficientAD-S. FastFlow ranks lower overall but remains competitive.

On the more challenging Rubber Mats dataset, EfficientAD-M and PatchCore-1% both reach 99.0% AU-ROC and 99.2% AU-PR, clearly outperforming EfficientAD-S. This indicates the importance of deeper feature representations for EfficientAD-M or larger backbones for PatchCore-1% when facing subtle, fine-grained defects.

At the category level, larger and visible defects, as in *leather* and *bottle*, are detected perfectly by all models. Differences emerge in smaller-defect categories, where EfficientAD-M leads on *screw* and *Rubber Mats*, PatchCore-1% also excels on *Rubber Mats*, and FastFlow performs best on *capsule* and *pill*.

The mean AU-ROC across datasets confirms EfficientAD-M as the top-performing model with 98.9%, and PatchCore-10% is very close (98.8%). EfficientAD-S performs slightly worse on Rubber Mats, reducing its mean score, while FastFlow achieves lower overall accuracy. AU-ROC classification error rates under F1-optimal thresholds range from 1.1% to 1.8%.

Furthermore, AU-PR results complement the AU-ROC findings and confirm strong true-positive detection of all models. Across datasets, PatchCore-10% achieves the highest mean AU-PR of 99.6%, closely followed by EfficientAD-M and PatchCore-1% (both 99.5%). All models usually achieve higher AU-PR than AU-ROC scores, especially on smaller-defect categories, suggesting reliable true anomaly detection but slight difficulties with false positive detections. These discrepancies illustrate that AU-PR is more sensitive to true positive detection in imbalanced or subtle-defect cases, whereas solely evaluating AU-ROC may underestimate performance, underlining the importance of considering both metrics for a reliable assessment.

Overall, EfficientAD-M provides the most stable accuracy across datasets, PatchCore-10% performs particularly well on object categories, and FastFlow shows its strengths in texture categories. The models' weakest categories, *pill* for PatchCore, *Rubber Mats* for EfficientAD-S, and *screw* for FastFlow, lower their average scores but also point to the specific use cases where model fine-tuning could be beneficial. No model dominates universally in every dataset category, which underscores the need for application-specific benchmarking.

#### 4.1.2 Efficiency Results

Efficiency results (see Tab. 2) reveal clear trade-offs. EfficientAD-S achieves the lowest latency of 4.5 ms and the highest throughput of 630 img/s, making it the fastest model by a large margin. EfficientAD-M offers a balanced profile with slightly higher latency of 6.8 ms but the best accuracy. PatchCore models achieve strong accuracy but are slower due to their memory bank design, while FastFlow shows moderate throughput with the highest latency. Thus, EfficientAD-S is optimal for high-speed applications, whereas EfficientAD-M and PatchCore-10% offer higher accuracy, but at increased computational cost.

#### 4.1.3 Discussion

The results highlight accuracy-efficiency trade-offs, illustrated in Fig. 2. *EfficientAD-M* achieves the most favorable balance of accuracy and speed, making it broadly suitable for industrial deployments. *PatchCore-10%* demonstrates comparable accuracy but at higher latency. *EfficientAD-S* is the fastest but shows reduced robustness on *Rubber Mats*. *FastFlow*, while overall slower and slightly less accurate, performs competitively in single smaller-defect and texture categories.

The differences in model performance can be partly explained by architectural design: *EfficientAD* benefits from its *Patch Description Network*, with the deeper M-variant providing improved feature representations. *PatchCore* also relies on patch-based embeddings and can benefit from using larger backbones, with coreset subsampling keeping the memory bank minimal and inference feasible. For instance, *PatchCore-1%* utilizes a larger Wide-ResNet101 and delivers higher accuracy for some smaller-defect categories such as *Rubber Mats* or *capsule*. Similarly, Roth et al. [11] achieve 99.6% AU-ROC on *MVTec AD* with *PatchCore-1% ensemble* and slightly higher resolution input, without major inference slowdowns. Nonetheless, the lower inference speed and increasing computational resource demands must be considered. With the strong performance of *EfficientAD* and *PatchCore*, we can conclude the effectiveness of utilizing patch-based feature representations for AD.

In summary, *EfficientAD-M* and *PatchCore-10%* deliver the best performance across datasets, *EfficientAD-S* provides unmatched efficiency, and *FastFlow* offers situational advantages. However, no single model is universally optimal for every specific scenario. Instead, tailored model selection based on use case-specific complexity, defect characteristics, and efficiency constraints is key to deploying AD systems effectively to the application context of industrial applications.

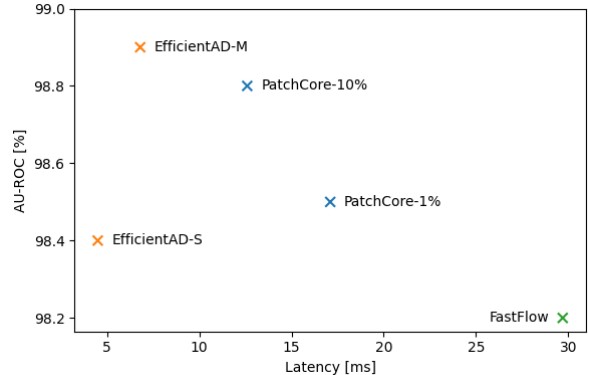

**Figure 2.** Accuracy-efficiency trade-off: mean AU-ROC [%] across datasets against inference latency [ms] of all models at image resolution 256×256.

### 4.2 Impact of Image Resolution

We next analyzed how input resolution influences model accuracy and efficiency. The experiment focused on categories containing smaller, fine-grained defects, where higher resolutions may help preserve important details lost through downsampling. At the same time, higher resolutions can increase computational demand, requiring a balance between potential accuracy gains and efficiency cost for practical viability.

#### 4.2.1 Accuracy Results

Fig. 3 shows the progression of accuracy across resolutions. Across MVTec AD, PatchCore exhibits the clearest positive relationship. Its accuracy rises steeply from 128×128 to 256×256, confirming the risk of excessive downsampling, and continues to improve at higher resolutions. Beyond the standard resolution, accuracy gains become smaller but remain consistent. At the largest resolution of 512×512, PatchCore-1% achieves the highest mean AU-ROC of 99.3% on MVTec AD, with the largest gains in smaller-defect categories such as *screw* and *pill*. Some categories, like *cable* and *capsule*, peak at the intermediate resolution of 320×320 before declining. This suggests that very high resolutions can introduce noise and redundancy, resulting in normal variation being mistaken for anomalies. On Rubber Mats, accuracy similarly peaks at the intermediate resolution and then decreases, showing that higher resolution is not universally beneficial.

EfficientAD-S, in contrast, performs best at the standard resolution input and generally declines with increasing resolution. There are minor improvements at moderately increased resolution in categories such as *capsule*, *cable*, and *Rubber Mats*, but they do not persist at larger sizes. This indicates that EfficientAD is optimized for standard-resolution input, and its lightweight architecture does not scale well to higher resolutions.

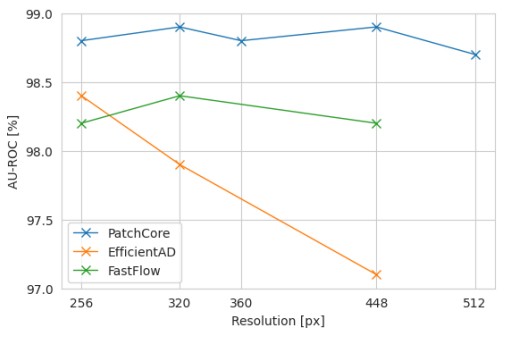

**(a)** Mean.

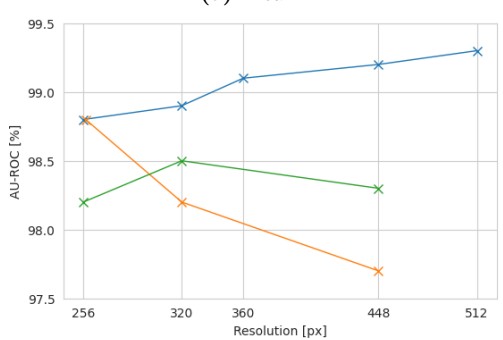

**(b)** MVTec AD.

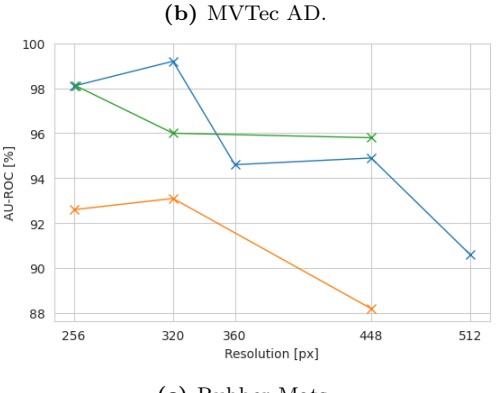

**(c)** Rubber Mats.

**Figure 3.** Progression of classification accuracy (AU-ROC [%]) for varying image resolutions on *MVTec AD*, *Rubber Mats*, and across both datasets.

FastFlow shows mixed, category-dependent behavior. For previously weaker categories such as *screw* and *toothbrush*, accuracy improves steadily with resolution, peaking at the largest size of 448×448. Other categories, like *pill* and *Rubber Mats*, achieve the best performance at standard or moderately increased resolutions but degrade at higher inputs. Overall, FastFlow benefits from intermediate resolution increases in some cases but does not follow a consistent trend across all categories.

In summary, moderately increased resolution can improve detection accuracy of smaller or previously complex defects, particularly for *PatchCore*, while *FastFlow* yields selective gains but with less consistent trends. *EfficientAD-S* generally suffers from reduced accuracy with increased resolution and is best kept at standard resolution.

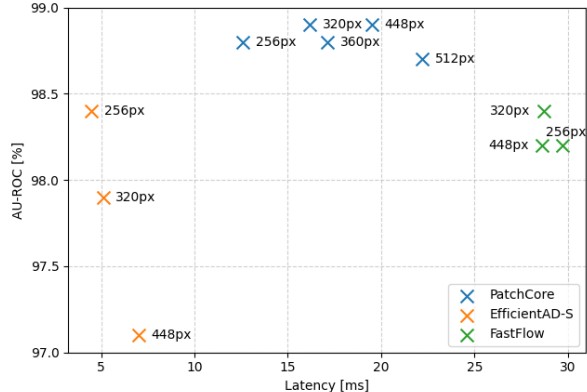

**Figure 4.** Accuracy-efficiency trade-off: mean AU-ROC [%] across datasets against inference latency [ms] of all models for varying image resolutions.

#### 4.2.2 Efficiency Results

Resolution scaling comes at the cost of rising computational demand, affecting inference efficiency. PatchCore is most impacted, where latency increases substantially with resolution due to its larger memory bank required to store more features. While the largest resolution yields the highest accuracy for MVTec AD, it also results in the slowest latency of 22.2 ms, highlighting a steep trade-off between accuracy and efficiency.

EfficientAD-S remains the most efficient model, with only slight increases in latency and minimal throughput reduction as resolution increases. Fast-Flow shows stable latency across resolutions, though it remains the slowest overall.

#### 4.2.3 Discussion

The accuracy-efficiency trade-off, illustrated in Fig. 4, is therefore highly model-dependent. Patch-Core can deliver meaningful accuracy improvements for subtle defects when resolution is increased, but with rising computational cost and efficiency loss. FastFlow offers stable efficiency and selective accuracy gains, whereas EfficientAD-S remains optimal at standard resolution and shows little benefit from scaling.

The declining accuracy of EfficientAD-S at higher resolutions can be explained by its lightweight student architecture, which is optimized for standard resolution inputs. At larger inputs, the student's receptive field becomes too small relative to the image size, leading to poor teacher-student alignment. While the teacher's features scale reasonably well, the student has a fixed architecture with shallow pooling, where the receptive field does not grow proportionally with higher resolutions, creating a mismatch between the teacher and student models. Additionally, the autoencoder upsamples based on image size, and larger resolutions can introduce

**Table 3.** Classification accuracy (AU-ROC/AU-PR, [%]) and inference efficiency (latency [ms]/throughput [img/s]/GPU memory [GB]) of the multi-class benchmark on *MVTec AD* with image resolution 256×256.

| Category | AD Model | AU-ROC | AU-PR | Latency | Throughput | Memory |
|----------|----------|--------|-------|---------|------------|--------|
| Textures | PatchCore | 98.8 | 99.6 | 30.7 | 47 | 16.7 |
| | EfficientAD | 96.8 | 99.0 | **4.6** | **630** | 3.2 |
| | FastFlow | **99.6** | **99.9** | 30.1 | 377 | 13.0 |
| Objects | PatchCore | 90.0 | 96.2 | 51.1 | 26 | 33.0 |
| | EfficientAD | 54.5 | 77.5 | **4.5** | **632** | 3.6 |
| | FastFlow | **91.1** | **96.4** | 30.7 | 376 | 13.0 |
| **Mean** | PatchCore | 94.9 | 97.9 | 41.1 | 37 | 24.9 |
| | EfficientAD | 75.7 | 88.3 | **4.6** | **631** | 3.4 |
| | FastFlow | **95.4** | **98.2** | 30.4 | 377 | 13.0 |

reconstruction noise.

Overall, the results show that resolution upscaling should be applied selectively. However, intermediate increases can improve accuracy for smaller-defect use cases. For applications prioritizing maximum accuracy on smaller defects, *PatchCore* with moderately higher resolutions can provide clear benefits, albeit with increased latency. For real-time or resource-constrained deployments, *EfficientAD-S* at standard resolution remains the most practical option. *Fast-Flow* sits in between, offering stable efficiency but only situational accuracy gains.

These findings emphasize that the impact of resolution depends on both model architecture and defect characteristics. While higher resolutions can improve sensitivity to fine details, diminishing returns or even accuracy loss occur beyond moderate upscaling. In practice, balancing resolution, accuracy, and efficiency is essential to optimize deployment.

## 4.3 Multi-Class Benchmark

We finally evaluated how models generalize when trained jointly across multiple *MVTec AD* categories grouped by textures and objects, rather than in separate single-class setups. This setting reflects a practical deployment scenario, where AD systems must handle diverse categories simultaneously. Results are summarized in Tab. 3, and a comparison to the single-class accuracy is included in Fig. 5 for reference.

### 4.3.1 Accuracy Results

Compared to the single-class benchmark, mean AU-ROC and AU-PR scores decline across all models. FastFlow and PatchCore-10% retain relatively strong performance with a moderate accuracy decrease of approximately 3-8%. In contrast, EfficientAD-S degrades sharply to 75.7% AU-ROC and 88.3% AU-PR, reflecting a substantial loss in robustness.

A closer breakdown reveals a consistent difference between textures and objects. On textures, model performance remains high and close to single-class

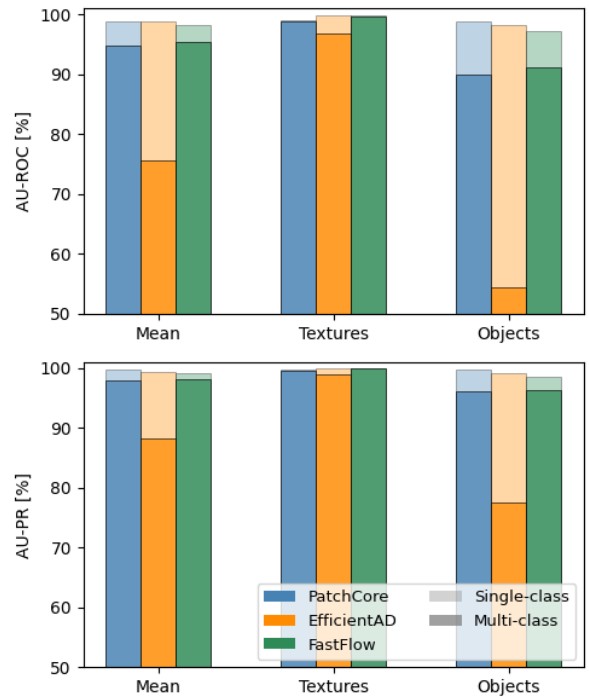

**Figure 5.** Accuracy comparison between single- and multi-class benchmarks on *MVTec AD* at 256×256 resolution. AU-ROC (top) and AU-PR (bottom) are shown, with single-class results overlaid transparently.

results. On objects, however, accuracy decreases more significantly. EfficientAD-S collapses to 54.5% AU-ROC and 77.5% AU-PR, while PatchCore-10% and FastFlow decline less, maintaining AU-ROC above 90% and AU-PR above 96%, respectively. This highlights the greater challenge posed by more heterogeneous object categories, where intra-class variability in normal samples makes separation from anomalies more difficult.

The divergence between AU-ROC and AU-PR offers further insight. Despite lower AU-ROC values, AU-PR remains comparatively high for PatchCore-10% and FastFlow, indicating that true anomalies are still detected reliably, but false positives increase under the multi-class setting. EfficientAD-S, however, suffers in both AU-ROC and AU-PR, revealing reduced sensitivity as well as higher false-positive rates.

### 4.3.2 Efficiency Results

Inference efficiency is largely unaffected for EfficientAD-S and FastFlow, which maintain the same latency and throughput as in the single-class benchmark. Both architectures benefit from their lightweight design, keeping GPU memory usage stable even for larger, multi-class datasets. The primary limitation of these models in this setting lies in accuracy rather than runtime.

PatchCore, by contrast, experiences substantial

efficiency degradation. Latency rises substantially, throughput decreases, and GPU memory usage grows sharply, as the memory bank must store many more feature representations from the larger multi-class training data. This scalability issue reduces the practicality of PatchCore for large datasets, as in multi-class deployments, when efficiency or resource constraints are critical.

### 4.3.3 Discussion

The multi-class benchmark demonstrates that generalization to heterogeneous data imposes a significantly greater challenge than in the single-class setting. While all models maintain strong performance on texture categories, object categories reveal weaknesses, with *EfficientAD-S* affected most severely. This difficulty of *EfficientAD* can be partly explained by its reconstruction-based autoencoder module, which can rather learn shortcuts (e.g., artificial noise elimination) and therefore fail to precisely reconstruct normal samples, resulting in false positive detections.

Among the evaluated models, *FastFlow* offers the most balanced trade-off, combining moderate accuracy decreases with stable efficiency. *PatchCore-10%* remains competitive in accuracy, but its growing memory and latency costs limit viability in resource-constrained scenarios. *EfficientAD-S* retains its efficiency advantage but struggles to generalize beyond homogeneous categories such as textures.

Overall, these findings indicate that multi-class AD remains challenging. While all models generalize well on texture categories, suggesting that multi-class AD on textures is practically viable, object categories expose performance gaps. The results underscore the importance of careful model selection depending on the diversity and scale of data expected in deployment.

## 5 Conclusion

This work presented a comprehensive benchmark of three modern unsupervised anomaly detection methods, namely *EfficientAD*, *PatchCore*, and *FastFlow*, on the *MVTec AD* and the new *Rubber Mats* dataset. The evaluation covers a single-class benchmark at standard resolution (see section 4.1), the impact of increased input resolution (see section 4.2), and a multi-class benchmark across grouped categories (see section 4.3). Model performance was analyzed jointly with accuracy and inference efficiency to reflect practical deployment scenarios.

In the standard resolution benchmark, *EfficientAD-M* achieved the best balance of accuracy and efficiency, combining high AU-ROC and AU-PR with low latency. *PatchCore* reached competitive accuracy, even on subtle or smaller-defect categories, but incurred higher computational costs and therefore slightly lower efficiency. *FastFlow* delivered stable efficiency and strong category-specific performance, though its overall mean accuracy was slightly lower.

Resolution scaling revealed that moderate upscaling can benefit smaller-defect categories, especially for *PatchCore* and in some cases *FastFlow*. However, accuracy gains diminish or even reverse at higher resolutions, and efficiency declines for *PatchCore*. *EfficientAD-S* particularly degraded with larger inputs, underscoring that resolution scaling must be applied selectively and in alignment with model design and use-case demands.

The multi-class benchmark highlighted the greater challenge of generalization to heterogeneous data. While all models maintained strong performance on texture categories, accuracy declined notably for objects. *PatchCore* and *FastFlow* preserved robust AU-PR scores and moderate accuracy decreases, while *EfficientAD-S* suffered a severe decline in both AU-ROC and AU-PR. These results confirm that multi-class anomaly detection remains more difficult, particularly for object categories with high intra-class variability.

In conclusion, the findings underline that no single model is universally optimal. *EfficientAD* is well-suited for real-time industrial applications that require both speed and accuracy. *PatchCore* offers high accuracy and can maximize detection accuracy in smaller-defect scenarios with higher resolutions, when efficiency is secondary and resources are sufficient. *FastFlow* provides a balanced alternative, combining competitive accuracy, robust multi-class generalization, and stable efficiency across resolutions and large datasets.

In order to come up with an answer for the initial research question (*"How can deep learning-based, unsupervised anomaly detection models for industrial image classification be best selected in practice?"*), we propose that effective anomaly detection in industrial imaging requires a context-aware model selection strategy that balances accuracy, efficiency, and generalization to the diversity of real-world data.

**Limitations and future work.** This study was restricted to two datasets and single training runs per configuration, which may obscure smaller performance variations. Future work should extend benchmarks to additional challenging datasets (e.g., MVTec LOCO, VisA), include further AD models, and evaluate repeated experiments and data augmentation strategies. Such extensions could provide deeper insights into generalization and robustness across industrial scenarios. Further, they would lead to a more versatile research answer.

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

# A   Appendix

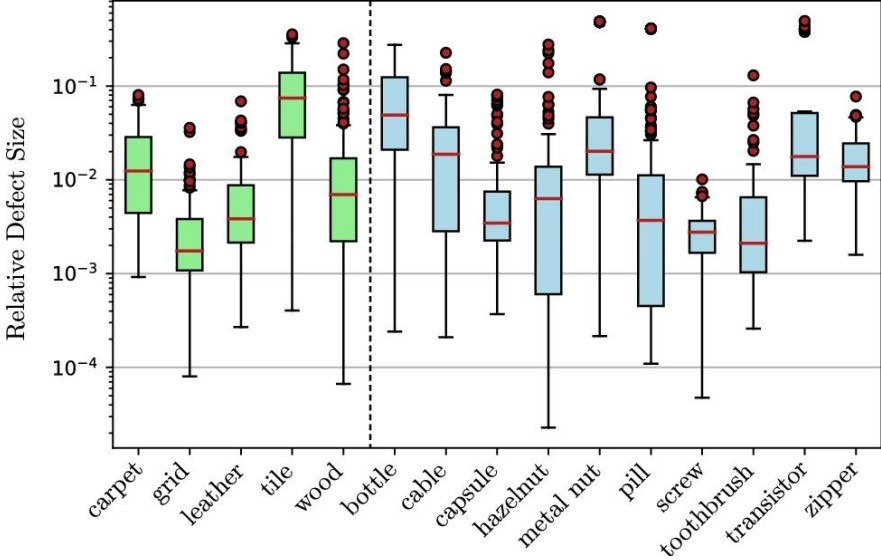

**Figure A.1.** Relative size of anomalies per MVTec AD categories as proposed by Bergmann et al. [8]. Textures (green) and objects (blue) are visualized in a box-whisker plot. Defect size is calculated by the size of anomalous pixels relative to the full image.

**Table A.1.** Model hyperparameters and settings for the experiments.

| Method | PatchCore | | EfficientAD | | FastFlow |
|---|---|---|---|---|---|
| | 10% | 1% | S | M | |
| Paradigm | Memory bank | | Teacher-student | | Distribution-map |
| Backbone | Wide-ResNet50 | Wide-ResNet101 | PDN | | Wide-ResNet50 |
| Resolution | 128/256 | 256/320/360/448/512 | 256/320/448 | | 256/320/448 |
| LR | / | | $10^{-4}$ | | $10^{-3}$ |
| Optimizer | / | | Adam | | Adam |
| Max | 1 epoch | | 70k steps | | 100 epochs |
| Specific | Subsampling=10% | Subsampling=1% | Small PDN | Medium PDN | Flow-steps=8 |

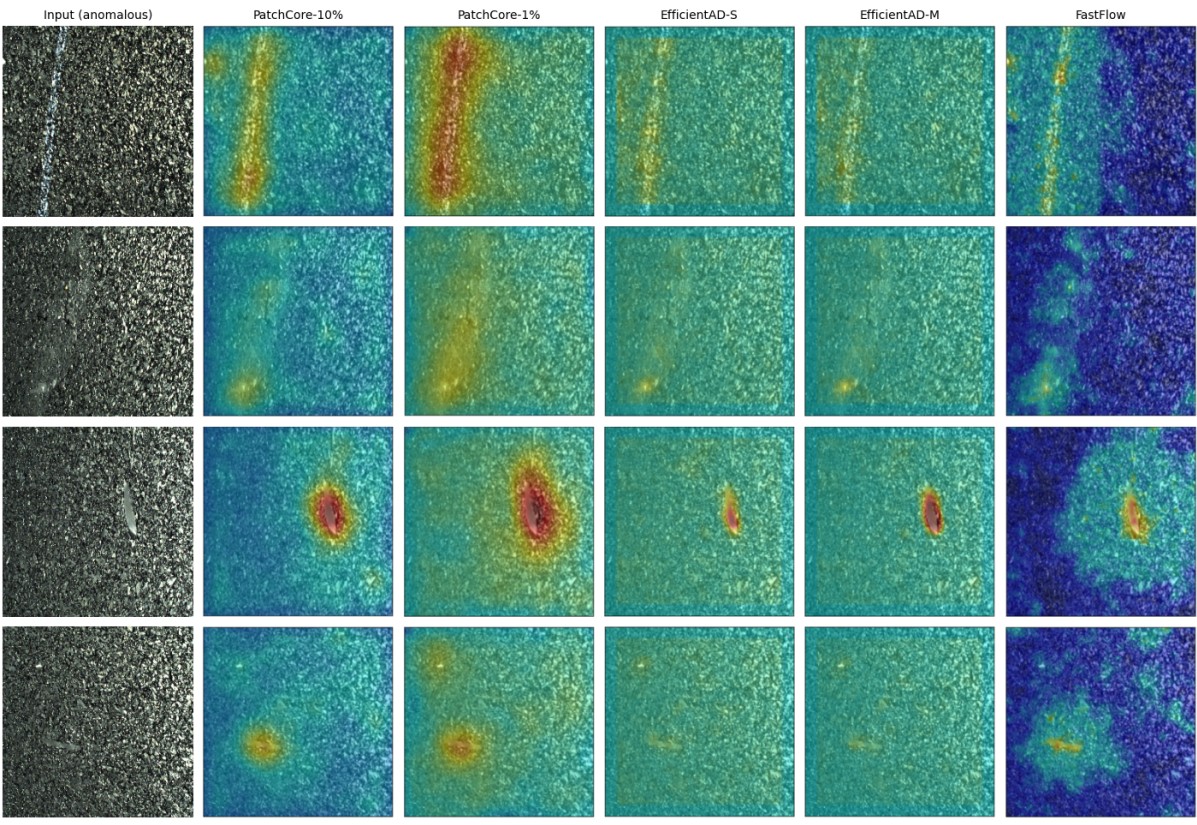

**Figure A.2.** Example anomaly heat maps for correct true positive detections on Rubber Mats. The top-performing models, PatchCore-1% and EfficientAD-M, highlight defects with higher anomaly scores compared to other models.

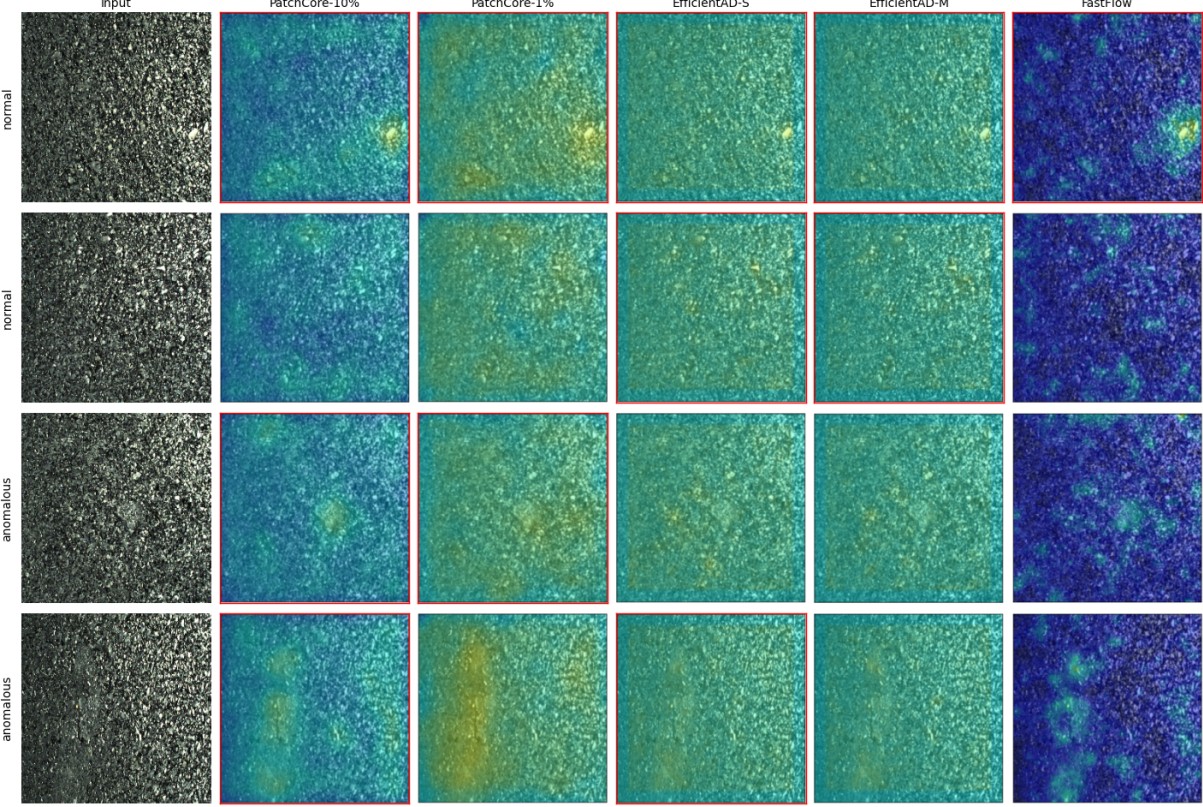

**Figure A.3.** Predicted anomaly heat maps for misclassified test images on Rubber Mats. The top two input images are normal and the bottom two are anomalous, while misclassified cases are highlighted with red borders. Notably, the top-row normal image is consistently detected as a false positive by all models.

