# OpenReview forum: "Deep Learning-Based Anomaly Detection in Industrial Images: Evaluation and Comparison of Modern Methods"
_NLDL.org/2026/Conference — Submitted to NLDL 2026_

### Official Review · Reviewer_9W3U · 2025-10-08
**Practically important paper**

**Rating:** 4
**Confidence:** 3
**Final Rating:** 4
**Final Confidence:** 4

**Summary:**

This paper considers the problem of unsupervised anomaly detection.
Authors conducted thorough experiments on a well-known benchmark dataset and a more realistic industry dataset.
Authors consider three methods and multiple problem settings, especially the multiclass generalization setting is of high pratical need.
Results can serve as a guide for selecting anomaly detection method based on the problem setting.

**Strengths:**

- This paper gives a comprehensive report on bechmarking different anomaly detection methods and different problem settings.
- The overall paper is well writtern and easy to follow.
- This can serve as a guide book for practioners to select methods based on the problem at hand.

**Weaknesses:**

- Besides the experimental findings that each method differs in stength, no deeper discussion regarding the design and model structure of each method is provided. This can give more insights for practioners who refer to this paper on selecting a proper method.

**Final Justification:**

The correctness and soundness of the paper is sufficient enough.

**Justification:**

Although being lack of theoretical insights, this paper is overall clearly written, well-structured and self-contained.

---

> ### Author Rebuttal · Authors · 2025-10-20
>
> First off, we want to thank the reviewer for the constructive feedback and for recognizing our paper as practically relevant with novelty worth sharing.
>
> ## Regarding the request for deeper model-level discussion:
>
> We agree that explaining the model performance differences based on architectural design provides valuable insight for practitioners. Our paper already offers such discussions, most notably in the Impact of Image Resolution (cf. section 4.2) and Multi-Class Benchmark (cf. section 4.3) sections. We attribute the observed model behaviors to architectural and implementation-based factors and phrase these as mechanistic explanations rather than proven causal claims.
>
> For example, EfficientAD’s lightweight student-teacher and autoencoder design relies on the student approximating teacher features and on the autoencoder/decoder reconstructing patch-level feature representations. As input size increases, the student’s capacity and the autoencoder’s reconstruction can become limiting factors, explaining the decline of EfficientAD-S at larger image resolutions (see lines 518-531 for upscaling behavior & lines 616-621 for multi-class behavior). PatchCore, for example, benefits from its memory-bank design, enabling robust generalization in multi-class AD while remaining reasonably efficient due to its coreset-subsampling. However, the memory bank size still increases sharply with larger datasets (which can be particularly observed in the multi-class setting) as more feature representations must be stored, which can reduce PatchCore’s practicality with large amounts of input data and high efficiency requirements (see lines 600-608).
>
> We appreciate this suggestion, and in a revised version, we are going to elaborate on this discussion further to more explicitly link each model’s structural properties to the observed trade-offs across benchmark settings.

---

### Official Review · Reviewer_znvi · 2025-10-10
**Review for Submission14**

**Rating:** 2
**Confidence:** 3

**Summary:**

This paper presents a benchmark comparison of 3 unsupervised anomaly detection methods on MVTec AD and a new Rubber Mats dataset. The authors evaluate their performance trade-offs on standard/higher resolutions and single/multi-class training settings to guide practical model selection for industrial deployment.

**Strengths:**

1. The paper evaluates models on multiple dimensions, and the authors provide practitioners with actionable insights.

2. The authors provide a real-world industrial dataset, the Rubber Mats dataset, which makes the evaluation more representative of actual deployment scenarios.

3. The focus on deployment-critical metrics enhances the practical utility of this work for industrial practitioners

**Weaknesses:**

1. This paper is empirical, benchmarking existing methods without proposing new techniques or architectural improvements.

2. The authors emphasize fine-grained defects and higher resolutions as key challenges, yet they only evaluate resolutions up to 512×512. The Rubber Mats dataset has a native resolution of 1440×1080 but is downsampled for experiments, contradicting their claims about the importance of preserving fine details.

3. Single training runs per configuration without confidence intervals or statistical significance testing make it impossible to distinguish actual performance differences from random variation. The absence of ablation studies fails to investigate what architectural components drive the observed performance differences.

4. The paper mentions F1-optimal thresholds (line 354). Still, it does not consistently apply this strategy across all experiments and fails to discuss how different operating points (precision-recall trade-offs) affect model selection for specific industrial QC requirements, where high recall may be prioritized over precision.

5. The preprocessing details (normalization strategies, augmentation techniques) are missing, and the threshold selection strategy for binary classification is not clearly described. Since different models may have different optimal thresholds, these omissions affect both reproducibility and the fairness of performance comparisons.

**Justification:**

While it provides a competent empirical comparison, it offers no new techniques, theoretical insights, or architectural improvements beyond benchmarking existing methods. More critically, there is a fundamental misalignment between claims and execution: the authors emphasize fine-grained defect detection at high resolution yet only evaluate up to 512×512 despite their dataset's native 1440×1080 resolution.

The paper may be read as an internal technical report rather than a research paper suitable for this venue.

---

> ### Author Rebuttal · Authors · 2025-10-20
>
> We thank the reviewer for the detailed and constructive feedback, as well as for recognizing the practical relevance and real-world dataset contribution of our work.
>
> ## (1) Scope and Contribution:
>
> We acknowledge that our study is rather empirical than proposing new architectures. However, our contribution lies in elaborating the systematical benchmark of unsupervised AD models under underexplored conditions (notably multi-class generalization and image-resolution scaling) and introducing the Rubber Mats dataset with fine-grained industrial defects. Recent benchmark papers (e.g., [6] Rolih et al. 2024 addressing resolution scaling and [10] Zhang et al., 2024 addressing multi-class AD) establish a comparable empirical focus. Further benchmarks, as we have presented in our rebuttal of reviewer #1, mention standard resolution settings (256x256) in benchmarks and are establishing a comparable empirical focus, too. However,  our study extends these research articles with (1) additional experimental settings and (2) a realistic dataset. Thus, benchmarks provided by our article broaden the applicability and present practical insights.
>
> ## (2) Image Resolution and Resource Constraints:
>
> We agree that the native resolution of Rubber Mats is higher than the tested sizes. However, we tested up to 512x512, which is double the common 256x256 input size and therefore a significant increase. Larger sizes quickly became infeasible given available compute: PatchCore’s memory bank scales with the number of patch embeddings (growing with input size), and EfficientAD’s student-teacher and autoencoder pipelines produce intermediate feature maps sized by the input, leading to large increases in GPU memory and training time. We did test larger input sizes initially but found them impractical for full systematic evaluation with all models. Nonetheless, the trends observed (saturation or decline in accuracy for PatchCore and EfficientAD-S, efficiency decline for PatchCore) are informative, suggesting that further upscaling would amplify these effects.
> However, we agree on your point. So, in a revised version, we are going to clarify the experiments mentioned and better justify this design decision by a documenting passage.
>
> ## (3) Multiple Runs and Variance:
> We agree that repeated runs could yield more robust statistics. So, we explicitly mentioned this in the Limitations section (cf. section 5) and clarified this to be future work. However, we have not intended to present an article having statistical analyses in its focus. We intend to present a design-oriented paper justifying further research about the dimensions observed. Here, the design and the practical application in real-world circumstances are in focus. Hopefully, you can appreciate its value.
>
> ## (4) Thresholds and Evaluation Metrics:
> We clarify that thresholding consistently follows Anomalib’s F1-adaptive strategy (F1-optimal with harmonic mean of precision and recall) across all models to ensure a consistent thresholding rule. We report AU-ROC and AU-PR as threshold-independent primary metrics; AU-PR is particularly informative for imbalanced AD scenarios. Since AU-ROC and AU-PR are threshold-independent, they more reliably reflect model separability across various thresholds. Thus, they allow for a broader model evaluation and are commonly used metrics for many AD benchmarks. However, precision-recall trade-offs for industrial quality control applications can be adapted by interpreting AU-PR results/curves and choosing thresholds according to specific needs (e.g., higher recall).
> We agree that this discussion can be expanded and will include a more detailed analysis of threshold effects in the revised article.
>
> ## (5) Preprocessing and Data Augmentation:
> All models use standard preprocessing pipelines implemented in Anomalib, following their respective original papers (e.g., ImageNet normalization and model-specific transforms per model implementation). No additional data augmentations were applied to maintain fair comparability, as stated in the methodology section (cf. section 3) and limitations (cf. section 5). However, as you are asking for these preprocessing details, we are going to present these details in a revision in a systematical way. We agree on the point that these are interesting for the readers.
>
> In summary, while our work focuses on benchmarking and practical real-world context rather than architectural innovation and sand-box testing, it (1) provides a comprehensive and practically oriented evaluation across models, (2) includes a new real-world dataset, and (3) evaluates multiple realistic deployment scenarios. Therefore, we aim to complement and extend prior research.

---

### Official Review · Reviewer_GnTN · 2025-10-10
**The authors propose a benchmark as to measure the performance of three kinds of anomaly detection models for defect identification in industrial settings. The research is well-organized, sound, and solid. The results aim to aid on decision-making weighting the methods by their discriminative capabilities (anomaly vs. no anomaly) and inference speed.**

**Rating:** 5
**Confidence:** 4

**Summary:**

The authors propose a benchmark to measure the performance of three kinds of anomaly detection models in industrial settings with the goal of assisting in decision-making when deciding which model would fit best particular anomaly detection needs. The research is well-organized, sound, and solid. The results show different methods are suitable when favouring discriminative performance vs. inference speed.

**Strengths:**

The authors have identified a particular problem (anomaly detection in industrial images) and have evaluated three distinct models to understand their discriminative performance and real-time inference. They compare three methods across a dataset and perform the comparisons across a variety of settings to understand the tradeoffs between the discriminative performance and the inference velocity.

**Weaknesses:**

We consider the paper to be solid. Nevertheless, we would like to point out certain improvement opportunities:

GENERAL COMMENTS:

(1)- We would appreciate the authors strengthening their related work by introducing a survey paper on the topic and grounding their choices regarding datasets and methods in the literature. The wider the adoption of the datasets and methods under consideration, the greater the comparability of the current research.

FIGURES

(2)- We encourage the authors to use a selection of colours that is friendly to colorblind people. Furthermore, the authors could use a monochromatic palette and distinct symbols for each series, ensuring differences can be quickly noticed.

(3)- The authors could introduce a spider web Figure to showcase how the different methods under consideration fare at different performance dimensions being considered.

TABLES

(4) - Align numbers to the right to ensure differences in magnitude become self-evident.

(5) - The authors should clarify what bolded results mean

MINOR COMMENTS

(6) - The term "Accuracy Results" may cause confusion, given that Accuracy is a particular metric associated with classification performance. We encourage the authors to find a different expression, e.g., "Model Discriminative Performance Results".

**Justification:**

The authors have written a very solid paper, creating a benchmark for anomaly detection in industrial images. The authors have proposed (i) datasets on which to execute the anomaly detection, (ii) models to be tested, and (iii) metrics used to assess the performance.

---

> ### Author Rebuttal · Authors · 2025-10-20
>
> We thank the reviewer for the positive and encouraging feedback, as well as the constructive suggestions for improvement opportunities that we will incorporate into the revised article. We address each comment below:
>
> ## (1) Strengthening Related Work:
>
> We appreciate this suggestion. Our selection of models (PatchCore, EfficientAD, and FastFlow) and datasets is indeed grounded in recent benchmark literature. PatchCore and FastFlow are among the most frequently adopted methods across recent AD benchmarks/surveys that we referenced in our work ([1] Cui et al., 2023; [2] Liu et al., 2024; [4] Batzner et al., 2024; [5] Tao et al., 2022; [6] Rolih et al., 2024; [10] Zhang et al., 2024; [13] Yu et al., 2021; [14] Xie et al., 2024; [17] Zheng et al., 2022; [19] Zou et al., 2022) and achieve among the best performances in terms of accuracy.
>
> These articles also include the widely adopted datasets such as MVTec AD [1, 2, 4, 5, 6, 9, 10, 11, 13, 14, 17], VisA [2, 4, 6, 10, 14, 19], and MVTec LOCO [2, 4, 9, 10, 14]. EfficientAD, while more recent, has shown very competitive (mostly even better) results compared to PatchCore and FastFlow ([4] Batzner et al., 2024) on MVTec AD, MVTec LOCO, and VisA. We also referenced benchmark studies with comparable experimental setups for our Image Resolution and Multi-class Benchmarks (e.g., [6] Rolih et al., 2024 (higher-resolution evaluation) and [10] Zhang et al., 2024 (multi-class benchmark)) to ensure comparability.
>
> MVTec AD remains a widely adopted dataset for unsupervised AD in numerous benchmarks, while we introduced the Rubber Mats dataset to extend evaluation toward more subtle, smaller-defect scenarios that can challenge recent AD methods.
>
> However, we agree on the idea of stating these references more clearly in our “Related Work” section. So, we are going to explicitly elaborate on these survey papers in a revised version of the article.
>
> ## (2-3) Figures:
>
> We agree that colorblind-friendly palettes are important. In a revised version, we are going to adopt a perceptually uniform, colorblind-safe colormap and add distinct markers to enhance clarity. We will also consider the reviewer’s suggestion of including a spider plot to visualize model trade-offs more intuitively.
>
> ## (4-5) Tables:
>
> We acknowledge these points entirely. So, in a revision, we are going to consider right-aligning numerical results. Further, we are going to explicitly clarify what bolded results mean; in captions we are going to state that bolded values indicate the best performances.
>
> ## (6) Terminology:
>
> We agree that “Accuracy Results” may not be accurate enough, and this term is often associated with classification performance. We thank you for your alternative, namely "Model Discriminative Performance Results", which will fit well. In a revision, we are going to revise the term currently used to increase reading clarity.

---

### Meta-Review · Area_Chair_ru3R · 2025-11-03

**Recommendation:** Reject
**Confidence:** 5

**Metareview:**

The initial assessment of the paper by the 3 reviewers led to a mixed opinion, and we thank the authors for the effort made in trying to answer reviewers' concerns in their rebuttal. Nevertheless, the shortcomings raised by one of the reviewer are valid and not properly addressed in the rebuttal. So while the paper has definitely some experimental value, the methodological flaws and the limited contribution (that I have also assessed after reading the paper) remain strong issues. Thus I recommend to reject the paper.

---

### Decision · Program_Chairs · 2025-11-05

**Decision:**

Reject

**Comment:**

Based on the reviewers and AC comments, the paper cannot be presented at the conference.